# Wild Patagonian yeast improve the evolutionary potential of novel interspecific hybrid strains for lager brewing

Jennifer Molinet[1,2,3]*, Juan P. Navarrete[1], Carlos A. Villarroel[1,4], Pablo Villarreal[1,2], Felipe I. Sandoval[1], Roberto F. Nespolo[1,5,6,7], Rike Stelkens[3], Francisco A. Cubillos[1,2,5]*

1 ANID-Millennium Science Initiative-Millennium Institute for Integrative Biology (iBio), Santiago, Chile, 2 Departamento de Biología, Facultad de Química y Biología, Universidad de Santiago de Chile, Santiago, Chile, 3 Department of Zoology, Stockholm University, Stockholm, Sweden, 4 Centro de Biotecnología de los Recursos Naturales (CENBio), Facultad de Ciencias Agrarias y Forestales, Universidad Católica del Maule, Talca, Chile, 5 ANID-Millennium Nucleus of Patagonian Limit of Life (LiLi), Valdivia, Chile, 6 Instituto de Ciencias Ambientales y Evolutivas, Facultad de Ciencias, Universidad Austral de Chile, Valdivia, Chile, 7 Center of Applied Ecology and Sustainability (CAPES), Santiago, Chile

* jennifer.molinet@usach.cl (JM); francisco.cubillos.r@usach.cl (FAC)

**Data Availability Statement:** All fastq sequences were deposited in the National Center for Biotechnology Information (NCBI) as a Sequence

## Abstract

Lager yeasts are limited to a few strains worldwide, imposing restrictions on flavour and aroma diversity and hindering our understanding of the complex evolutionary mechanisms during yeast domestication. The recent finding of diverse *S. eubayanus* lineages from Patagonia offers potential for generating new lager yeasts with different flavour profiles. Here, we leverage the natural genetic diversity of *S. eubayanus* and expand the lager yeast repertoire by including three distinct Patagonian *S. eubayanus* lineages. We used experimental evolution and selection on desirable traits to enhance the fermentation profiles of novel *S. cerevisiae* x *S. eubayanus* hybrids. Our analyses reveal an intricate interplay of pre-existing diversity, selection on species-specific mitochondria, *de-novo* mutations, and gene copy variations in sugar metabolism genes, resulting in high ethanol production and unique aroma profiles. Hybrids with *S. eubayanus* mitochondria exhibited greater evolutionary potential and superior fitness post-evolution, analogous to commercial lager hybrids. Using genome-wide screens of the parental subgenomes, we identified genetic changes in *IRA2*, *IMA1*, and *MALX* genes that influence maltose metabolism, and increase glycolytic flux and sugar consumption in the evolved hybrids. Functional validation and transcriptome analyses confirmed increased maltose-related gene expression, influencing greater maltotriose consumption in evolved hybrids. This study demonstrates the potential for generating industrially viable lager yeast hybrids from wild Patagonian strains. Our hybridization, evolution, and mitochondrial selection approach produced hybrids with high fermentation capacity and expands lager beer brewing options.

Read Archive under the BioProject accession number PRJNA1043100 (http://www.ncbi.nlm.nih.gov/bioproject/104100).

**Funding:** This research was funded by Agencia Nacional de Investigación y Desarrollo (ANID) FONDECYT program and ANID-Programa Iniciativa Científica Milenio – ICN17_022 and NCN2021_050. FAC is supported by FONDECYT grant N° 1220026, JM by FONDECYT POSTDOCTORADO grant N° 3200545 and PV by ANID FONDECYT POSTDOCTORADO grant N° 3200575. CAV is supported by FONDECYT INICIACIÓN grant N° 11230724. RFN is supported by FONDECYT grant N° 1221073. RS and JM's work is supported by the Swedish Research Council (2022-03427) and the Knut and Alice Wallenberg Foundation (2017.0163). FIS, JM, and PV received a salary from FONDECYT grant N° 1220026, FONDECYT POSTDOCTORADO grants N° 3200545 and N° 3200575, respectively. The funders had no role in study design, data collection and analysis, decision to publish, or preparation of the manuscript.

**Competing interests:** The authors have declared that no competing interests exist.

## Author summary

Lager beer dominates the global market, accounting for over 90% of commercial beer varieties. The main player in lager fermentation is the yeast *Saccharomyces pastorianus*, an interspecific hybrid between *S. cerevisiae* and *S. eubayanus*. Despite its popularity, the range of flavours and aromas found in lager beers is restricted by the low genetic diversity of available lager strains. Here, we explored if lager yeast profiles can be diversified by leveraging natural isolates of *S. eubayanus* from Chilean Patagonia. We generated *de novo* hybrids between *S. cerevisiae* and three distinct *S. eubayanus* Patagonian lineages. Through experimental evolution and selection on fermentation traits, we improved the fermentation profiles of the hybrids. We found that mutations in *IRA2*, *IMA1*, and *MALX* genes enhanced their maltose and maltotriose metabolism, resulting in higher ethanol production and unique aroma profiles. Our results also confirm that *S. eubayanus* mitochondria confer a greater evolutionary potential than *S. cerevisiae* mitochondria. The current study encourages the use of wild yeast strains to develop new brewing applications to expand the repertoire of *de novo* lager yeasts.

## Introduction

Humans have paved the way for microbes, such as yeast, to evolve desirable features for making bread, wine, beer, and many other fermented beverages for millennia [1]. The fermentation environment, characterized by limited oxygen, high ethanol concentrations, and microbial competition for nutrients (typically yeasts, molds, and bacteria) can be considered stressful [2]. One evolutionary mechanism to overcome harsh conditions is hybridization, because it rapidly combines beneficial phenotypic features of distantly related species and generates large amounts of genetic variation available for natural selection to act on [3–5]. Hybrids can also express unique phenotypic traits not seen in the parental populations through the recombination of parental genetic material, enabling them to thrive in different ecological niches [4,6–8]. An iconic example is the domesticated hybrid yeast *Saccharomyces pastorianus* to produce modern lager (pilsner) beers. *S. pastorianus* results from the successful interspecies hybridization between *S. cerevisiae* and *S. eubayanus* [9,10]. Hybrids have been shown to benefit from the cold tolerance of *S. eubayanus* and the superior fermentation kinetics of *S. cerevisiae* [11]. We now know that domestication over the last 500 years has generated lager yeast strains with the unique ability to rapidly ferment at lower temperatures resulting in a crisp flavour profile and efficient sedimentation, improving the clarity of the final product. However, the genetic diversity of commercial lager yeast strains is extremely limited, mainly due to the standardization of industrial lager production during the nineteenth century in Germany [9,12]. This gave rise to only two genetically distinct *S. pastorianus* subgroups, Group 1 strains ('Saaz') and Group 2 strains ('Frohberg'). The poor genetic diversity of lager strains used in commercial brewing today (85 lager strains commercially available versus 358 ale strains [13]) puts tight constraints on the variety of flavours and aromas found in lager beer. At the same time, it limits our understanding of the evolutionary mechanism operating during the yeast domestication process.

The discovery of *S. eubayanus* in Patagonia in 2011 [14], opened new possibilities for creating novel hybrid strains by using the full range of natural genetic diversity found in this species. Phylogenetic analyses have revealed six distinct lineages of *S. eubayanus*, including China, Patagonia A ('PA'), Holarctic, and Patagonia B, 'PB-1', 'PB-2' and 'PB-3', and some admixed strains derived from ancient crosses [15,16]. Of these, *S. eubayanus* from Patagonia displays

the broadest phenotypic diversity for a wide range of traits, including high maltose consumption, aroma profiles, and fermentation capacity [15,17,18]. The distinctive traits of wild Patagonian *S. eubayanus* strains indicate their potential for crafting new lager beer styles. These strains could yield novel taste and aroma profiles, approaching similar complexity and diversity in flavour, appearance, and mouthfeel as Ale beers.

Lager yeast hybrids experienced an intense domestication process through selection and re-pitching during beer fermentation since the 17th century [9,10,12,19,20], a process similar to experimental evolution [21,22]. Experimental evolution with microbes is a powerful tool to study adaptive responses to selection under environmental constraints [23–26]. Recent studies on novel *S. cerevisiae* x *S. eubayanus* hybrids suggest that hybrid fermentative vigour at low temperature results from a variety of genetic changes, including loss of heterozygosity (LOH), ectopic recombination, transcriptional rewiring, selection of superior parental alleles [27], heterozygote advantage due to the complementation of loss-of-function mutations in the F1 hybrid genome [28], and novel structural and single nucleotide variants in the hybrid genome [29]. A recent transcriptome analysis of a laboratory-made lager hybrid strain under fermentation conditions highlighted that the regulatory 'cross-talk' between the parental subgenomes caused a novel sugar consumption phenotype in the hybrid (maltotriose utilization, essential for lager fermentation), which was absent in both parental strains [28]. Although these studies have greatly contributed to our understanding of the genetic basis of different lager phenotypes, most studies only considered a single *S. eubayanus* genetic background (type strain CBS 12357), which alone is not representative of the species-rich genetic diversity.

Here, we hybridized three different *S. cerevisiae* and *S. eubayanus* strains to generate genetically and phenotypically diverse novel lager hybrids via spore-to-spore mating. The initial *de novo* hybrids had fermentation capacities comparable to those of their parental strains and did not show positive heterosis. However, when we subjected hybrids to a 'fast motion' improvement process using experimental evolution under different fermentation conditions for 250 generations, they exceeded the fitness of the ancestral hybrids, particularly those retaining the *S. eubayanus* mitochondria. Superior hybrid fitness was explained by faster fermentation performance and greater maltose and maltotriose consumption We found that copy number variation in *MAL* genes in the *S. cerevisiae* subgenome, together with SNPs in genes related to glycolytic flux, induced significantly greater expression levels of *MAL* and *IMA1* genes, and led to improved fitness under fermentative conditions in these novel *S. cerevisiae* x *S. eubayanus* yeast hybrids. Furthermore, evolved hybrids had significantly distinct aroma profiles, varying significantly from the established profiles found in lager beer.

## Materials and methods

### Parental strains

Three *S. cerevisiae* strains were selected for hybridization from a collection of 15 strains isolated from different wine-producing areas in Central Chile and previously described by Martinez et al. [30]. Similarly, three *S. eubayanus* parental strains were selected from a collection of strains isolated from different locations in Chilean Patagonia, exhibiting high fermentative capacity and representative of the different Patagonia-B lineages (PB-1, PB-2, and PB-3) [15]. The *S. pastorianus* Saflager W34/70 (Fermentis, France) strain was used as a commercial lager fermentation control. All strains were maintained in YPD agar (1% yeast extract, 2% peptone, 2% glucose and 2% agar) and stored at -80˚C in 20% glycerol stocks. Strains are listed in Tab A in **S1 Table**.

## Interspecific hybrids strains and mitochondria genotyping

Parental strains were sporulated on 2% potassium acetate agar plates (2% agar) for at least seven days at 20°C. Interspecific F1 hybrids were generated through spore-spore mating between *S. eubayanus* strains and *S. cerevisiae* strains (**S1 Fig**). For this, tetrads were treated with 10 µL Zymolyase 100 T (50 mg/mL) and spores of opposite species were dissected and placed next to each other on a YPD agar plates using a SporePlay micromanipulator (Singer Instruments, UK). Plates were incubated at two different temperatures, 12 and 20°C, for 2–5 days to preserve the cold- and heat-tolerant mitochondria, respectively, as previously described [31,32], resulting in nine different F1 hybrids (ranging from H1 until H9, Tab A in **S1 Table**). This procedure was repeated on 25 tetrads of each species, for each type of cross (H1 to H9) and temperature (12 and 20°C), resulting in 18 different cross x temperature combinations. Finally, colonies were isolated, re-streaked on fresh YPD agar plates, and continued to be incubated at 12 and 20°C. The hybrid status of isolated colonies was confirmed by amplification of rDNA-PCR (ITS1, 5.8S, and ITS2) using universal fungal primers ITS1 and ITS4 [33], followed by digestion of the amplicon using the *Hae*III restriction enzyme (Promega, USA) as previously described [34] on one colony for each cross attempt (**S1 Fig**). Confirmed F1 hybrids were designated as H1 to H9 based on parental strains, followed by the hybridization temperature (12 or 20) and the colony number (i.e. H1.20–1 depicts cross 1 at 20°C (Tab A in **S1 Table**)). We identified the mitochondrial genotype by Sanger sequencing the mitochondrial *COX3* gene as previously described [32].

## Beer wort fermentation and metabolite screening

Fermentations were carried out in three biological replicates using previously oxygenated (15 mg/L) 12°P wort, supplemented with 0.3 ppm $ZnCl_2$ as previously described [17]. Briefly, pre-cultures were grown in 5 mL 6°P wort for 24 h at 20°C with constant agitation at 150 rpm. Cells were then transferred to 50 mL 12°P wort and incubated for 24 h at 20°C with constant agitation at 150 rpm. Cells were collected by centrifugation and used to calculate the final cell concentration to inoculate the subsequent fermentation according to the formula described by White and Zainasheff [35]. Cells were inoculated into 50 mL 12°P wort in 250 mL bottles covered by airlocks containing 30% glycerol. The fermentations were incubated at 12 or 20°C, with no agitation for 15 days and monitored by weighing the bottles daily to determine weight loss over time.

Sugar (glucose, fructose, maltose and maltotriose) consumption and ethanol production were determined by High-Performance Liquid Chromatography (HPLC) after 14 days of fermentation. Filtered samples (20 µL) were injected in a Shimadzu Prominence HPLC (Shimadzu, USA) with a BioRad HPX-87H column using 5 mM sulfuric acid and 4 mL acetonitrile per liter of sulfuric acid as the mobile phase at a 0.5 mL/min flow rate. Volatile compound production was determined by using HeadSpace Solid-Phase MicroExtraction followed by Gas Chromatography-Mass Spectrometry (HS-SPME-GC/MS) after 14 days of fermentation as previously described [18].

## Phenotypic characterization

Hybrids and parental strains were phenotypically characterized under microculture conditions as previously described [36]. Briefly, we estimated mitotic growth in 96-well plates containing Yeast Nitrogen Base (YNB) supplemented with 2% glucose, 2% maltose, 2% maltotriose, 2% glucose and 9% ethanol, 2% glucose and 10% sorbitol, and under carbon source switching (diauxic shift) from glucose to maltose as previously described [37]. All conditions were evaluated at 25°C. Lag phase, growth efficiency, and the maximum specific growth rate ($\mu_{max}$) were

determined as previously described [38,39]. For the diauxic shift between glucose and maltose, lag time and $\mu_{max}$ were determined during growth in maltose. The parameters were calculated following curve fitting (OD values were transformed to ln) using the Gompertz function [40] in R (version 4.03).

Mid-parent and best-parent heterosis were determined as previously described [41,42], using Eq 1 and 2, where mid-parent heterosis denotes the hybrid deviation from the mid-parent performance and best-parent heterosis denotes the hybrid deviation from the better parent phenotypic value [43].

$$Mid-parent\ heterosis = \frac{Phenotypic\ value_h}{Phenotypic\ value_p} \qquad (1)$$

$$Best-parent\ heterosis = \frac{Phenotypic\ value_h}{Phenotypic\ value_{bp}} \qquad (2)$$

Where:

$$Phenotypic\ value_h = phenotypic\ value_{hybrid}$$

$$Phenotypic\ value_p = \frac{phenotypic\ value_{parental1} + phenotypic\ value_{parental2}}{2}$$

$$Phenotypic\ value_{bp} = max(phenotypic\ value_{parental\ 1}, phenotypic\ value_{parental2})$$

## Experimental evolution

Experimental evolution was carried out at 20°C under two different media conditions (M and T): 1) YNB + 2% maltose supplemented with 9% ethanol (M) and 2) YNB + 1% maltose + 1% maltotriose supplemented with 9% ethanol (T). Experimental evolution assays in maltose were performed in a final volume of 1 mL in 2 mL tubes, while those in maltose and maltotriose were performed in a 96-well plate under a final volume of 200 μL. Each hybrid strain was first grown in 0.67% YNB medium with 2% maltose at 25°C for 24 h with constant agitation at 150 rpm. Each pre-inoculum was then used to inoculate each evolution line at an initial $OD_{600nm}$ of 0.1, with three replicate lines per strain in medium M and four replicate lines in medium T. Lines in medium M were incubated at 20°C for 72 h. Lines in medium T were incubated for 144 h at 20°C. After this, cultures were then serially transferred into fresh medium for an initial $OD_{600nm}$ of 0.1. Serial transfers were repeated for 250 generations in total (approximately seven months). The number of generations was determined using the formula log(final cells–initial cells)/$log_2$ [44]. Population samples were stored at -80°C in 20% glycerol stocks after 50, 100, 150, 200 and 250 generations. After 250 generations, three colonies were isolated for each replicate line on YPM solid medium (1% yeast extract, 2% peptone, 2% maltose and 2% agar) supplemented with 6% ethanol. The fastest growing colonies were stored at -80°C in 20% glycerol stocks. The fitness increase of each the 28 evolved line was determined as the ratio between the phenotypic value of a given line and the equivalent of its respective ancestral hybrid.

## Genomic characterization

Genomic DNA was obtained for whole-genome sequencing using the YeaStar Genomic DNA Kit (Zymo Research, USA) and sequenced in an Illumina NextSeq500 following the manufacturer's instructions. Variant calling and filtering were done with GATK version 4.3.0.0 [45].

Briefly, cleaned reads were mapped to a concatenated reference genome consisting of *S. cerevisiae* strain DBVPG6765 [46] and S. *eubayanus* strain CL216.1 [17] using BWA mem 0.7.17 [47], after which output bam files were sorted and indexed using Samtools 1.13 [48]. Variants were called per sample using HaplotypeCaller (default settings) generating g.vcf files. Variant databases were built using GenomicsDBImport and genotypes were called using GenotypeGVCFs (-G StandardAnnotation). SNPs and INDELs were extracted and filtered out separately using SelectVariants. We then applied recommended filters with the following options: QD < 2.0, FS > 60.0, MQ < 40.0, SOR > 4.0, MQRankSum < -12.5, ReadPosRankSum < -8.0. This vcf file was further filtered by removing missing data using the option–max-missing 1, filtering out sites with a coverage below 5th or above the 95th coverage sample percentile using the options–min-meanDP and–max-meanDP, and minimum site quality of 30 (—minQ 30) in vcftools 0.1.16 [49]. Sites with a mappability less than 1 calculated by GenMap 1.3.0 [50] were filtered using bedtools 2.18 [51]. As an additional filtering step, the ancestral and evolved vcf files were intersected using BCFtools 1.3.1 [52] and variants with shared positions were extracted from the vcf files of the evolved hybrids. Annotation and effect prediction of the variants were performed with SnpEff [53].

We used sppIDer [54] to assess the proportional genomic contribution of each species to the nuclear and mitochondrial genomes in each sequenced hybrid. In addition, we used the tool to identify potential aneuploidies within these genomes. CNVs were called using CNVkit (—method wgs,—-target-avg-size 1000) [55]. As the analysis was performed on a haploid reference (both parental genomes were present), a CNV of log2 = 1 corresponds to a duplication.

Since *S. cerevisiae* and *S. eubayanus* show divergence greater than 20% we could not detect loss of heterozygosity (LOH) by mapping to a single parental reference. As both hybrids appear to have one copy of each parental genome with no aneuploidies, LOH should appear as a loss of coverage segments that can be detected as copy number losses. LOH regions were detected by mapping reads to the concatenated genome and we used CNVkit [55] to detect genomic segments (1000 bp windows) showing CNVs with log2FC less than -2. Comparisons were made between parental strains vs ancestral hybrids and between ancestral hybrids with evolved hybrids.

## RNA-seq analysis

Gene expression analysis was performed on ancestral and evolved hybrid strains H3-A and H3-E. RNA was obtained and processed after 24 h under beer wort fermentation in triplicates, using the E.Z.N.A Total RNA kit I (OMEGA) as previously described [37,56]. Total RNA was recovered using the RNA Clean and Concentrator Kit (Zymo Research). RNA integrity was confirmed using a Fragment Analyzer (Agilent). Illumina sequencing was performed in NextSeq500 platform.

Reads quality was evaluated using the fastqc tool (https://www.bioinformatics.babraham.ac.uk/projects/fastqc/) and processed using fastp (-3 l 40) [57]. Reads were mapped to a concatenated fasta file of the DBVPG6765 and CL216.1 genome sequences. To account for mapping bias due to the different genetic distances of the parental strains to their reference strains, the L3 and CL710.1 parental strains were re-sequenced using WGS, after which genomic reads were mapped with BWA [47] to the DBVPG6765 and CL216 references and SNPs were called using freebayes [58]. These SNPs were used to correct the hybrid genome sequence using the GATK FastaAlternateReferenceMaker tool. RNAseq reads were mapped to this hybrid reference using STAR (-outSAMmultNmax 1, -outMultimapperOrder random) [59]. Counts were obtained with featureCounts using a concatenated annotation file [60]. Counts were further analyzed in R using de DESeq package [61]. A PCA analysis to evaluate the

reproducibility of replicates was performed, after which two outlier replicates (H3-A replicate 3 and H3-E replicate 2) were removed. To analyze differences in allele expression, a list of 1-to-1 orthologous genes between both parental strains was identified using OMA [62]. Orthologous genes that differ more than 5% on their gene lengths were excluded. The differential allelic expression of these orthologous genes was determined using design = ~parental, with parental being "L3" or "CL710". Furthermore, orthologous genes that showed differential allele expression depending on the ancestral or evolved strain background were assessed using an interaction term (~ parental:condition), with condition being "ancestral" or "evolved". Finally, to evaluate differences between ancestral and evolved hybrid strains, all 11,047 hybrid genes (5,508 *S. eubayanus* and 5,539 *S. cerevisiae*) were individually tested for differential expression using DESeq2. Overall gene expression differences were evaluated using the design ~condition. For all analyzes an FDR < 0.05 was used to consider statistical differences. GO term enrichment analyzes on differentially expressed genes were calculated using the package TOPGO [63].

### *IRA2* gene validation

The *S. cerevisiae IRA2* polymorphism was validated by Sanger sequencing. PCR products were purified and sequenced by KIGene, Karolinska Institutet (Sweden). The presence of the SNP in the evolved hybrid strains was checked by visual inspection of the electropherograms. Null mutants for the *IRA2* gene in the *S. cerevisiae* subgenome were generated using CRISPR-Cas9 [64] as previously described [36]. Briefly, the gRNAs were designed using the Benchling online tool (https://www.benchling.com/) and cloned into the pAEF5 plasmid [65], using standard "Golden Gate Assembly" [66]. Ancestral and evolved hybrids were co-transformed with the pAEF5 plasmid carrying the gRNA and the Cas9 gene, together with a double-stranded DNA fragment (donor DNA). The donor DNA contained nourseothricin (NAT) resistance cassette, obtained from the pAG25 plasmid (Addgene plasmid #35121), flanked with sequences of the target allele, corresponding to 50-pb upstream of start codon and 50-pb downstream of the stop codon. Correct gene deletion was confirmed by standard colony PCR. All primers, gRNAs, and donor DNA are listed in Tab B in **S1 Table**.

### FACS analysis

DNA content was assessed through the propidium iodide (PI) staining assay, as previously described [67]. Initially, cells were recovered from glycerol stocks on YPD solid media and allowed to grow overnight at 25˚C. Subsequently, a portion of each patch was transferred into liquid YPD media and incubated overnight at 25˚C. Then, 1 ml of each culture was harvested and suspended in 2.3 ml of cold 70% ethanol for fixation during 48h h at 4˚C. Following fixation, cells were washed with sodium citrate (50 mM, pH 7) and 100 μl of cells resuspended in the same solution, and 1 μL of RNAse A (100 mg/mL) were incubated for 2 h at 37˚C. Then, cells were stained with a solution containing PI (final concentration of 50 μg/mL) and sodium citrate (50 mM, pH 7), and incubated for 40 min at room temperature in darkness. Analysis was conducted on a BD FACSCanto II flow cytometer with excitation at 488 nm and fluorescence collection using an FL2-A filter, analyzing ten thousand cells per sample. Three strains with known ploidy (two *S. cerevisiae* -n and 2n- and one *S. pastorianus* -4n-) were employed as controls.

### Statistical analysis

Data visualization and statistical analyses were performed with R software version 4.03. Maximum specific growth rates and total $CO_2$ loss were compared using an analysis of variance

(ANOVA) and differences between the mean values of three replicates were tested using Student's t-test and corrected for multiple comparisons using the Benjamini-Hochberg method. A *p-value* less than 0.05 ($p < 0.05$) was considered statistically significant. Heatmaps were generated using the ComplexHeatmap package version 2.6.2. A principal component analysis (PCA) was performed on phenotypic data using the FactoMineR package version 2.4 and the factoextra package version 1.07 for extracting, visualizing and interpreting the results.

## Results

### *De novo S. cerevisiae* x *S. eubayanus* F1 hybrids show similar phenotypes as their parental strains

The *S. cerevisiae* and *S. eubayanus* parental strains were selected from a previously described collection of Chilean isolates by Martinez et al. [30] and Nespolo, Villarroel et al. [15], respectively (Tab A in **S1 Table**). Initially, three *S. cerevisiae* strains from vineyards were selected because they showed: i) the highest maximum $CO_2$ loss in beer wort (**S2A Fig** **and S2 Table**), ii) the best growth performance under maltotriose conditions (**S2B Fig**), and iii) the most efficient maltotriose uptake during microculture conditions (**S2C Fig**). These strains were L3, L270, and L348. The selection of *S. eubayanus* parental strains was determined by two criteria: i) to represent distinct lineages found in the Chilean Patagonia to maximize genetic diversity (one strain per lineage, PB-1, PB-2, and PB-3), and ii) to display the highest $CO_2$ loss during fermentation when compared to strains within their respective lineages based on previous assays [15]. In this way, we selected CL450.1, CL710.1 and CL216.1, from PB-1, PB-2, and PB-3, respectively. All strains were able to sporulate (Tab C in **S2 Table**).

Nine interspecific F1 hybrid crosses were performed through spore-to-spore mating at 12°C and 20°C, to promote the preservation of the cold- and heat-tolerant mitochondria, respectively, as previously described (**S1 Fig**) [31,32]. We obtained 31 interspecific hybrids (Tab A in **S1 Table**), which we phenotyped individually under microculture conditions resembling those encountered during beer wort fermentation, such as glucose, maltose, maltotriose, and in the presence of ethanol and simulating osmotic stress with sorbitol (**S3 Table**). Hierarchical clustering of the phenotypic data denotes three main clusters, where there was no discernible clustering of hybrids based on their parental strains or hybridization temperature, highlighting the considerable phenotypic diversity resulting from hybridization (**Fig 1A**). To describe the phenotypic landscape of the 31 hybrids more comprehensively, we conducted a PCA analysis (**Fig 1B**). The individual factor map shows that hybrids made at 20°C fall into the right upper quarter of the phenotype space, and are associated with a higher growth rate in media with maltose and glucose compared to hybrids made at 12°C. This was particularly the case for four hybrid strains (H1, H3, H4 and H6), involving parental strains L3, L270, CL216.1 and CL710.1 (all p-values < 0.05, one-way ANOVA, Tab B in **S3 Table**).

To assess the impact of hybridization on yeast fitness, we calculated best-parent and mid-parent heterosis coefficients across the 31 hybrids (**Fig 1C** **and** Tabs C and D in **S3 Table**). While some hybrids exhibited positive mid-parent heterosis in 5 out 7 conditions (Tab C in **S3 Table**), we generally did not observe hybrids with positive best-parent heterosis (BPH, Tab D in **S3 Table**), except for rare cases involving maltose utilization and growth rate during diauxic shift, where 2 and 5 hybrids, respectively, displayed positive values (**Fig 1C**). Overall, inter-species hybridization did not result in a significant enhancement of fitness in F1 hybrids.

Next, we assessed the fermentation capacity of the 31 hybrids in wort at low temperature (**Fig 1D** **and S4 Table**). Hybrids generated at 12°C displayed similar levels of $CO_2$ production compared to those obtained at 20°C (**Fig 1D** **and** Tab A in **S4 Table,** p-value = 0.17, one-way ANOVA). We did not observe any hybrids exhibiting superior fermentative capacity when

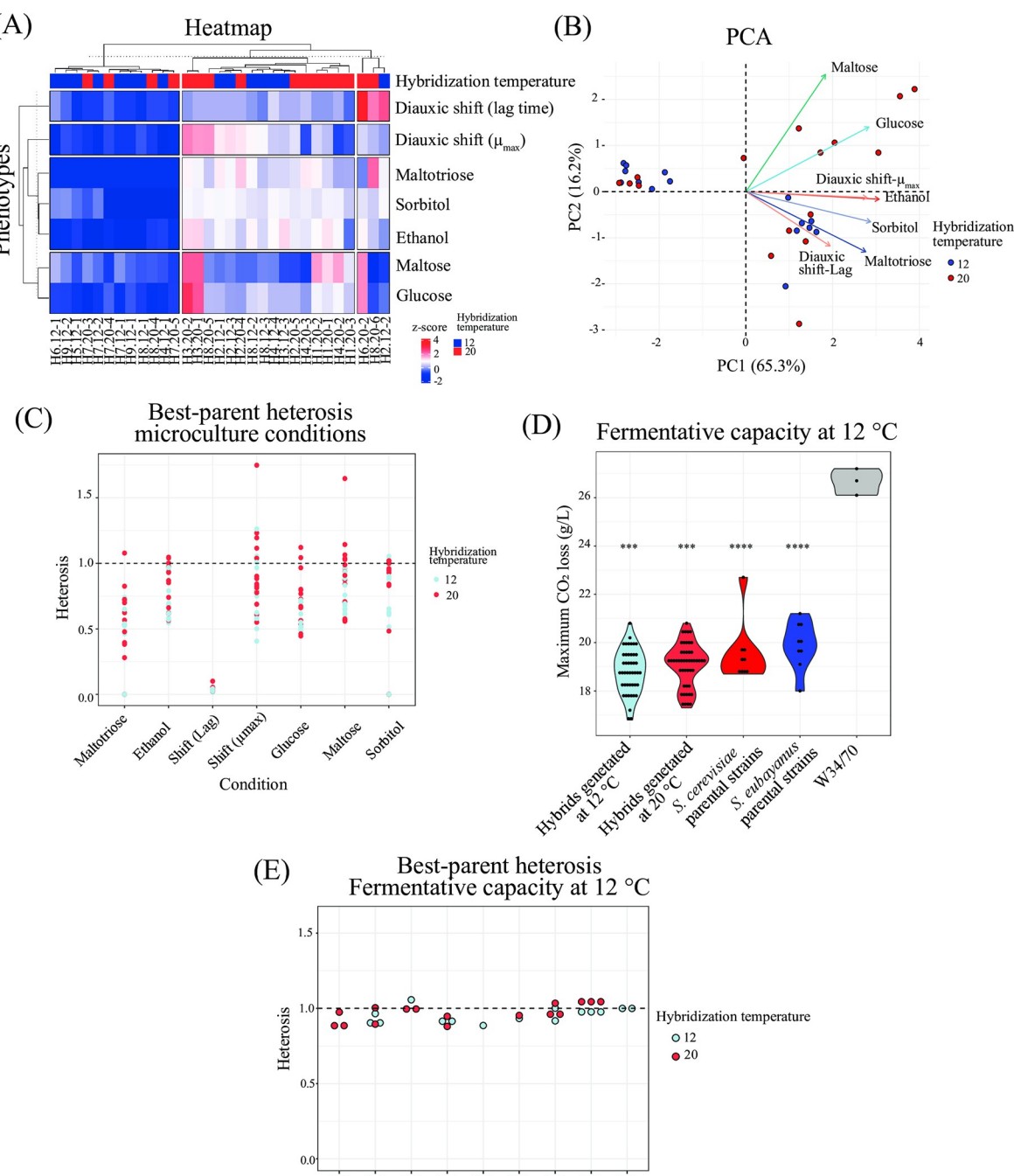

**Fig 1. Phenotypic characterization of interspecific F1 hybrids.** A) Hierarchically clustered heatmap of phenotypic diversity of 31 interspecific hybrid strains under microculture conditions. Phenotypic values are calculated as normalized z-scores. For the diauxic shift between glucose and maltose, lag time and $\mu_{max}$ were determined during growth in maltose. (B) Principal component analysis (PCA) using the maximum specific growth rates under six microculture growth conditions, together with the distribution of hybrid strains. Arrows depict the different environmental conditions. (C) Best-parent heterosis in the 31 interspecific hybrids evaluated under microculture conditions in triplicates. (D) Fermentation capacity for the 31 interspecific hybrids and parental strains at 12˚C. Plotted values correspond to mean values of three independent replicates for each hybrid. Asterisk indicates different levels of significance compared to the commercial strain W34/70 (Student t-test; *** p≤ 0.001 and **** p≤ 0.0001). (E) Best-parent heterosis in the 31 interspecific hybrids evaluated under fermentation conditions at 12˚C.

compared to their respective parental strains (**S3 Fig**), and there was no evidence for hybrid vigour according to best-parent and mid-parent heterosis coefficients (**Fig 1E and** Tabs C and D in **S4 Table**). Neither parents nor hybrids reached the fermentative capacity of the commercial strain W34/70 (p-value < 0.05, one-way ANOVA).

### Evolved lines carrying the *S. eubayanus* mitochondria exhibit a greater fitness under fermentation

All results so far indicated that the *de novo* interspecific hybrids did not show any hybrid vigour, in none of the phenotypes assessed. We thus decided to subject hybrids to experimental evolution to enhance their fermentative capacity. We specifically selected four hybrids (H3.12–3, H4.12–4, H6.20–2, and H8.20–5) because they demonstrated the highest phenotypic values across kinetic parameters. From here on we will refer to these strains as H3-A, H4-A, H6-A, and H8-A (A for 'ancestral' or unevolved hybrid). These four hybrids completely consumed the sugars present in the beer wort, except for maltotriose, which may explain the lower fermentative capacity of the hybrids compared to the commercial strain W34/70 (Tab E in **S4 Table**). Furthermore, these four hybrids represent crosses made at 12°C and 20°C and they encompass all six parental genetic backgrounds. To enhance the fermentative capacity of these selected hybrids, they were subjected to adaptive evolution at 20°C for 250 generations under two distinct conditions: i) YNB supplemented with 2% maltose and 9% ethanol (referred to as "M" medium), and ii) YNB supplemented with 1% maltose, 1% maltotriose, and 9% ethanol (referred to as "T" medium). We evolved three lines independently per cross in medium M, and four independent lines per cross in medium T. These conditions were chosen because maltose is the main sugar in beer wort (approximately 60%) [68]. Considering that yeast typically consume carbon sources in a specific order (glucose, fructose, maltose, and maltotriose), we employed a combination of maltose and maltotriose to facilitate the utilization of the latter carbon source.

After 250 generations, the evolved lines showed different levels of fitness improvements, depending on the environmental conditions and their genetic background (**Figs 2A and S4**), with distinct fitness trajectories over time (**S5 Fig**). All evolved lines significantly increased in fitness in at least one of the evolution media and/or kinetic parameters assessed compared to their respective ancestral hybrids (**Fig 2A and** Tabs A and B in **S5 Table**; p-value < 0.05, one-way ANOVA). Interestingly, evolved lines from hybrids made at 12°C mating temperature (H3-A and H4-A) showed a more pronounced fitness increase in the T medium compared to those generated at 20°C (p-value = 3.327e-08, one-way ANOVA, **Figs 2B** and **S4B**), suggesting that hybrids with *S. eubayanus* mitochondria have greater potential for improvement than hybrids with *S. cerevisiae* mitochondria. We verified that the two ancestral H3-A and H4-A hybrids carried only *S. eubayanus* mitochondria by sequencing the *COX3* gene, while H6-A and H8-A inherited the mitochondria from *S. cerevisiae* (Tabs C and D in **S5 Table**).

Next, we assessed the fermentative capacity of the evolved lines under conditions resembling beer wort fermentation (12°Brix and 12°C) (**Figs 2C and S6** and Tabs A and B in **S6 Table**). We did not observe a significant increase in $CO_2$ production levels in the evolved lines of the H6-A and H8-A hybrids in either M or T media (**Figs 2C and S6** and Tab B in **S6 Table**, p-value < 0.05, one-way ANOVA). However, we found a significant greater $CO_2$ production in the evolved lines of H4-A, evident in both evolution media, indicative of higher fermentation activity. The evolved lines of H3-A under T media also demonstrated a slightly higher $CO_2$ production (**Fig 2C** and Tab B in **S6 Table**, p-value < 0.05, one-way ANOVA, for H4 evolved lines and p-values of 0.0708 and 0.05149 for H3 evolved lines in M and T, respectively). Thus, both evolved hybrid lines generated at cold-temperature, carrying *S. eubayanus*

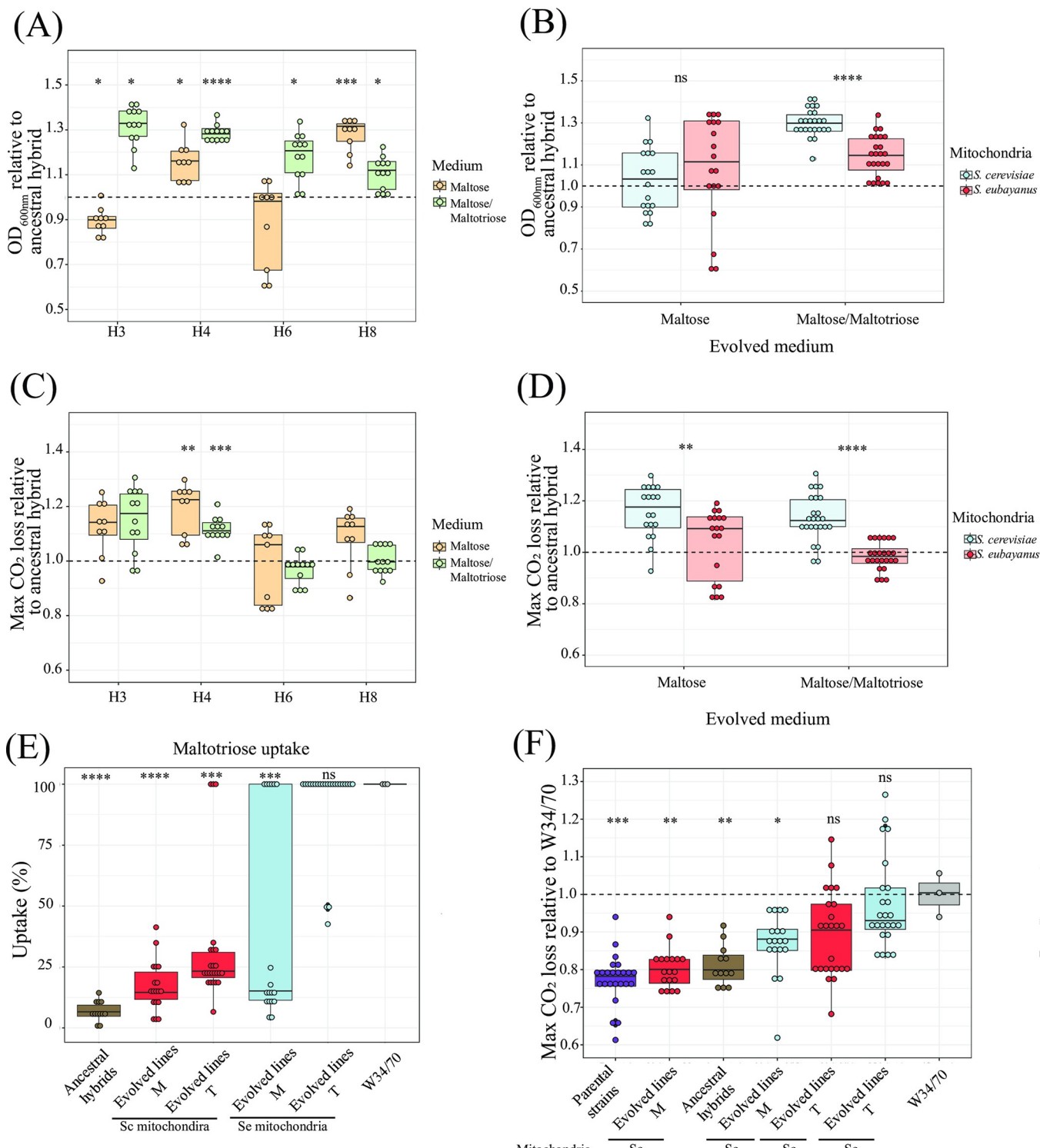

**Fig 2. Fitness of evolved lines under microcultures and fermentation conditions.** (A) Mean relative fitness (maximum $OD_{600nm}$) of evolved lines after 250 generations to their respective ancestral hybrids under microculture conditions. Evolved lines were evaluated in the same medium where they were evolved (M or T medium). (B) Comparison of mean relative fitness (maximum $OD_{600nm}$) shown in (A) between evolved lines from hybrids with *S. eubayanus* (Se) and *S. cerevisiae* (Sc) mitochondria. (C) Mean relative fitness (maximum $CO_2$ loss) of evolved lines after 250 generations to their respective ancestral hybrids under fermentation conditions at 12˚C. (D) Comparison of mean relative fitness (maximum $CO_2$ loss) shown in (C) between evolved lines from hybrids with *S. eubayanus* and *S. cerevisiae* mitochondria. (E) Maltotriose uptake of evolved hybrid lines in maltose (M) and maltose/maltotriose (T), relative to the commercial lager strain W34/70. Ancestral hybrids are shown in grey, and hybrid lines with *S. eubayanus* and *S. cerevisiae* mitochondria are shown in blue and

red, respectively. (F) The fermentative capacity of evolved individuals relative to the commercial lager strain W34/70 grouped according to the environmental condition used during experimental evolution and inherited mitochondria. Plotted values correspond to the mean of three independent biological replicates of each evolved line or strain. Asterisk indicates significant statistical differences between evolved lines and their respective ancestral hybrids in (A) and (C), between evolved lines with different inherited mitochondria in (B) and (D), and between evolved lines and the commercial lager strain in (E) and (F). Purple depicts Parental strains, brown the ancestral hybrid, and red and blue the Sc and Se evolved lines carrying mitochondria, respectively. Asterisk represents different levels of significance (Students t-test, * $p \leq 0.05$, ** $p \leq 0.01$, *** $p \leq 0.001$, **** $p \leq 0.0001$, ns not significant).

mitochondria, showed a greater increase in $CO_2$ production than hybrids carrying the *S. cerevisiae* mitochondria (**Fig 2C**). Specifically, hybrids with *S. eubayanus* mitochondria increased their maximum $CO_2$ loss by 10.6% when evolving in M medium (p-value = 0.003698, one-way ANOVA) and by 13% in T medium (p-value = 1.328e-08, one-way ANOVA) (**Fig 2D**). This was predominantly due to an elevated maltotriose uptake (**Fig 2E** and Tab C in **S6 Table**). Notably, the fermentative capacity of these hybrids reached that of the commercial strain (Tab D in **S6 Table**, p-value > 0.05, one-way ANOVA). These findings strongly suggest that lines derived from hybrids generated at colder temperatures carrying *S. eubayanus* mitochondria and evolved in a complex maltose/maltotriose medium (T), significantly enhanced their lager fermentative capacity due to an increased maltotriose uptake during beer wort fermentation.

## Isolation of evolved genotypes with improved fermentative capacity and maltotriose uptake

To isolate individual representatives from the evolved population lines, we obtained one single genotype from each of the four hybrid lines at 250 generations (28 genotypes in total), which were then subjected to phenotypic evaluation in beer wort. These individual genotypes showed similar fermentation profiles as the population-level analyses above (**Fig 2F**). Evolved hybrid genotypes carrying *S. eubayanus* mitochondria and evolved in T medium (maltose/maltotriose, H3-E and H4-E), showed higher $CO_2$ production compared to H6-E and H8-E (p-value < 0.05, ANOVA, **S7 Fig**). The genotypes with the largest significant fitness increase were derived from lines H3-3 and H3-4 evolved in T conditions (**S7 Fig**), which exceeded the commercial strain. Interestingly, two genotypes deriving from H6-A (carrying the *S. cerevisiae* mitochondria) evolved in T medium also showed a $CO_2$ loss similar to the commercial strain (p-value = 0.90372, one-way ANOVA).

To focus more in-depth on the evolved lines with the highest fermentative capacity and carrying the *S. eubayanus* mitochondria (H3-4 and H4-1 evolved in T medium, **S7 Fig**), we isolated three colonies from each of these two lines to evaluate their fermentative capabilities. Notably, the $CO_2$ loss kinetics among these genotypes were comparable (p-value > 0.05, one-way ANOVA), with genotype #1 from line H3-4 exhibiting the highest $CO_2$ loss (**Fig 3A**). All these genotypes' fermentation profiles closely resembled that of the W34/70 commercial lager strain, underscoring the significantly high fermentative capacity of these novel hybrids (p-value > 0.05, one-way ANOVA, **Fig 3A**). All genotypes consumed the maltotriose in the T medium completely (Tab A in **S7 Table**), and ethanol production ranged from 3.50% to 3.78% v/v (**Fig 3B**), which is similar to the commercial strain (p-value > 0.05, one-way ANOVA). One genotype (H4-1-C3) showed a remarkable 7.1% increase in ethanol production compared to the commercial strain (**Fig 3B**, p-value = 0.001, one-way ANOVA).

To compare the aroma profile of the H3-4-C1evolved hybrid to the lager strain, we identified volatile compounds (VCs) by HS-SPME-GC-MS in the fermented wort. This assay allowed us to identify 15 and 14 compounds in the evolved and commercial lager strains, respectively. We observed significant differences for 11 different compounds (**Fig 3C**, p-value < 0.05, one-way ANOVA, Tab B in **S7 Table**), including ethyl esters and higher alcohols.

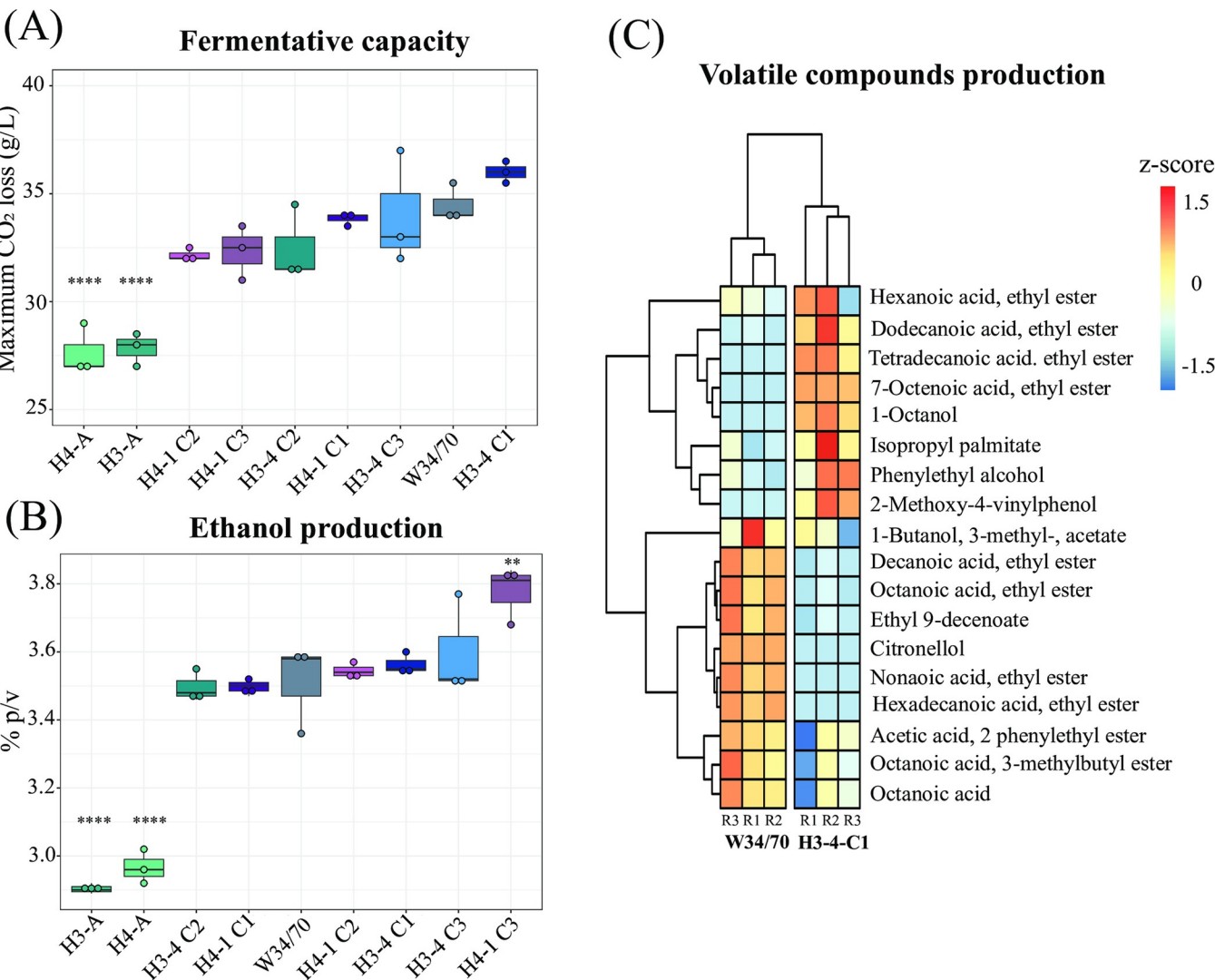

**Fig 3. Fermentation performance of evolved hybrid individuals.** (A) Maximum $CO_2$ loss (g/L) for three different isolated genotypes (C1-C3) from evolved lines H3-4 and H4-1, ancestral hybrids (H3-A and H4-A) and commercial lager strain (W34/70). (B) Ethanol production (% v/v) for strains evaluated in (A). (C) Hierarchically clustered heatmap of volatile compounds production for strains evaluated in (A). Phenotypic values are calculated as normalized z-scores. For (A) and (B), plotted values correspond to the mean of three independent replicates. The (*) represents different levels of significance between hybrids and commercial lager strains (Student t-test, ** $p < 0.01$, **** $p < 0.0001$).

For example, the evolved strain produced significantly more fatty acid ethyl esters, such as dodecanoic and tetradecanoic acid ethyl esters (p-value = 0.013 and 0.002, respectively, one-way ANOVA), which could provide sweetish notes. The commercial lager strain on the other hand produced higher amounts of other fatty acid ethyl esters, such as octanoic acid and nonanoic acid (**Fig 3D**, p-value = 0.001 and 0.0001, respectively, one-way ANOVA), and citronellol (p-value < 0.0001, one-way ANOVA), which is associated with a citrus aroma. The H3-4-C1evolved hybrid produced detectable levels of 4-vinphenol, associated with clove-like, spicy, and phenolic aromas, commonly found in Belgian ale and wheat beers, which was completely absent in the lager strain. These results demonstrate that the aroma profiles of the evolved hybrid differ from the commercial lager strain. Therefore, the beers produced with the evolved hybrid would have a different profile towards a more herbal, spicy, and phenolic

character. In contrast, those produced with the commercial strain would have a more citrusy and refreshing profile.

## Mutation in *IRA2* affects fermentation capacity in the evolved hybrid

To identify mutations in evolved hybrids associated with their improved fermentative capacity, we sequenced the genomes of the two genotypes exhibiting the highest $CO_2$ production levels, specifically H3-4-C1 and H4-1-C1 (from here on referred to as H3-E and H4-E; with 'E' for evolved hybrid) that were evolved in the maltose/maltotriose T medium (Tab A in **S8 Table**). Genome sequencing revealed that these two strain backgrounds had equal contributions from both parental genomes and that they had euploid, diploid genomes with no detectable aneuploidies (Tab B in **S8 Table**), containing the *S. eubayanus* mitochondria. We only detected small LOH regions that were associated with the hybridization process, mainly losses of *S. eubayanus* DNA at subtelomeric regions (Tab C in **S8 Table**) but also a small LOH region partly spanning the *HKR1* gene located in chromosome II, between the H4-A hybrid compared with the CL216.1 parent (Tab C in **S8 Table**).

We then identified *de novo* single nucleotide polymorphisms (SNPs) in the evolved hybrid genomes that were absent in the ancestral hybrids. We found 3 and 4 SNPs in the H3-E and H4-E backgrounds, respectively (Tab D in **S8 Table**). The evolved hybrids presented a similar number of SNPs per genome. In H3-E, we found 1 and 2 SNPs in the *S. cerevisiae* and *S. eubayanus* parental genomes, respectively, while H4-E had 1 and 3 SNPs in the corresponding parental genomes. We identified an anticipated stop-codon in the *IRA2* allele (encoding for a GTPase-activating protein, **Fig 4A**) in the *S. cerevisiae* subgenome, and a missense mutation in *CKB2* (encoding for a subunit of casein kinase 2) and *CMC1* (encoding for a mitochondrial protein necessary for full assembly of Cytochrome c oxidase) in the *S. eubayanus* subgenome (Tab D in **S8 Table**) in H3-E. While in H4-E, we identified a missense mutation in *MDS3* (a putative component of the TOR regulatory pathway), *FSF1* (predicted to be an alpha-isopropyl malate carrier), and *ASC1* (core component of the small ribosomal subunit) in the *S. eubayanus* subgenome, and an upstream gene mutation in *SSK2* (MAP kinase kinase kinase of HOG signalling pathway) in the *S. cerevisiae* subgenome (Tab D in **S8 Table**).

To detect additional genetic changes not identified in the individual clones and to track the relative frequencies of the *de novo* mutations in the evolution lines, we sequenced whole population samples at increasing time points of experimental evolution (at 50, 100, 150, 200, and 250 generations; **S8 Fig** **and** Tabs D and E in **S8 Table**). In this way, we identified 1 and 2 additional variants in the H3 and H4 evolved lines, respectively. For example, in the H4-evolved lines, we identified mutations within the *SCC2* and *NSE1* coding regions of two genes related to DNA replication and repair processes (Tab D in **S8 Table**). Subsequently, we estimated the *IRA2* mutation frequencies on the H3-evolved line. The *IRA2*-L2418* polymorphism arose before generation 50 and was completely fixed by 150 generations. We also determined the ploidy of this population at the end of the evolution process (250 generations), confirming that ploidy levels did not change during the experimental evolution process in the H3 population, maintaining a diploid state (**S9 Fig**).

To determine the phenotypic impact of the stop-codon detected in the *IRA2* gene, we performed a CRISPR-Cas9 gene editing targeting the *S. cerevisiae IRA2*, generating null mutants ($ira2^{Sc}$) in the evolved and non-evolved hybrids. We evaluated growth under microculture conditions in the same evolutionary medium (T) and under beer wort fermentation (**Fig 4B**). This assay revealed that $ira2^{Sc}$ mutants in the H3-A hybrid background had a 12.5% lower $OD_{max}$ under maltose/maltotriose conditions compared to H3-A (**Fig 4B**, p-value = 0.01213, one-way ANOVA), but still a similar fermentative capacity (**Fig 4C**, p-value = 0.79685, one-

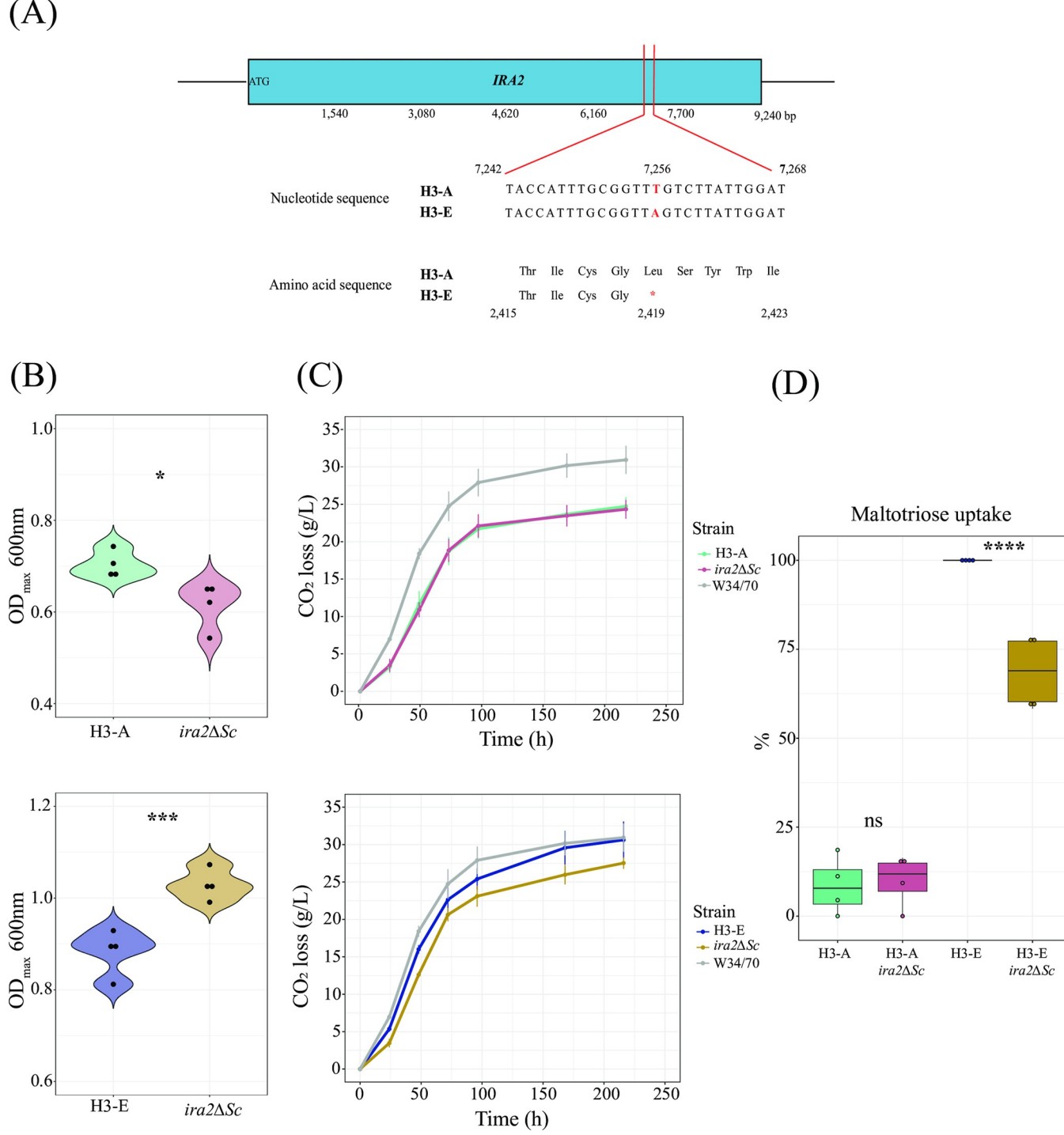

**Fig 4. Genomic analysis of evolved hybrids.** (A) SNP present in the *IRA2* gene in the *S. cerevisiae* subgenome in the H3-E hybrid. (B) Maximum $OD_{600nm}$ of *ira2Δ$^{Sc}$* mutant strains under microculture conditions. Mutant and wild-type strains were evaluated in the T medium. (C) $CO_2$ loss kinetics for *ira2Δ$^{Sc}$* mutant and wild-type strains. (D) Maltotriose uptake (%) for strains evaluated in (C). For (B), (C), and (D), plotted values correspond to the mean of four independent replicates. The (*) represents different levels of significance between mutant and wild-type strains (Student t-test, * $p < 0.05$, *** $p < 0.001$, **** $p < 0.0001$).

way ANOVA). In the H3-E hybrid, the null $ira2^{Sc}$ mutant showed a 16.6% higher $OD_{max}$ under microculture conditions (**Fig 4B**, p-value = 0.00042, one-way ANOVA) and a significantly lower fermentative capacity under beer wort, with a 13.6% decrease in $CO_2$ production (**Fig 4C**, p-value = 0.02315 one-way ANOVA, **S9 Table**) and a 10.8% decrease in the maximum $CO_2$ loss rate (**S9 Table**, p-value = 0.02268 one-way ANOVA). This decrease in the fermentative capacity in the H3-E null mutant correlates with a lower maltotriose uptake (68.6%, **Fig 4D**). These results suggest that the stop-codon in *IRA2* in the evolved hybrids does not necessarily lead to a loss of protein function, but instead to a complex genetic interaction in the H3-E background promoting a trade-off between biomass and fermentative capacity, which is likely partly responsible for the phenotypic differences during the evolutionary process.

## Copy number variants of genes related to maltose metabolism are associated with improved fermentative capacity in evolved hybrids

Since *ira2* null mutants did not restore the full increase in fermentative capacity of the evolved hybrids, we examined genes exhibiting copy number variation (CNVs) in H3-E and H4-E hybrids (**Fig 5A** and Tab F in **S8 Table**). Both H3-E and H4-E hybrids contained changes in copy number, particularly in the *MAL* gene family (**Fig 5A** and Tab F in **S8 Table**). For example, we identified 2 and 4 extra copies of the *MAL13* and *MAL11* genes in H4-E and H3-E, respectively.

To determine the impact of these mutations and the CNVs in the transcriptome of the H3-E hybrid, we estimated transcript abundance under beer fermentative conditions in the evolved and non-evolved hybrid. We identified 40 Differentially Expressed Genes (DEGs, FDR < 5%, Tab G in **S8 Table**), where 21 and 19 genes were up- and down-regulated in the evolved hybrid relative to its hybrid ancestor, respectively. Interestingly, we found that *S. cerevisiae* alleles for *IMA1*, *MAL11*, and *MAL13* were up-regulated in H3-E, which correlates with the increased gene copy number (**Fig 5B**). A GO term analysis showed that genes involved in maltose metabolic processes were up-regulated and genes in cell wall organization were down-regulated in the evolved hybrid (Tab H in **S8 Table**).

To measure the impact of *cis*-variants on allelic expression within each parental subgenome, we estimated allele specific expression (ASE) in the evolved and non-evolved hybrids (**Fig 5C** and Tab I in **S8 Table**). Seven genes showed ASE differences between the evolved and ancestral hybrid, likely originating from mutations in regulatory regions acquired during experimental evolution (**Fig 5C** and Tab I in **S8 Table**). Of these, one and six ASE differences in the H3-E hybrid represented up-regulated alleles in the *S. cerevisiae* and *S. eubayanus* subgenomes, respectively (Tab I in **S8 Table**). Interestingly, we detected the up-regulation of the *REG2* allele related to sugar consumption with a 2.1 higher fold change in the *S. cerevisiae* subgenome, which is involved in the regulation of glucose-repressible genes (**Fig 5C**), correlating with the higher maltose and maltotriose consumption levels in the evolved hybrid.

## Discussion

The hybrid yeast strains traditionally used for lager beer production (*S. pastorianus*) are highly limited in genetic diversity. Currently, only two types of strains are used worldwide [9,10,13,19], stemming from a single hybridization event that gave rise to the current lager strains. This strongly constrains the diversity of available flavour and aroma profiles. The genetically depleted landscape of lager strains also prevents a comprehensive understanding of the genetic changes crucial for the adaptation process [9,10]. The recent discovery of genetically and phenotypically distinct *S. eubayanus* lineages, including isolates from Patagonia [14–

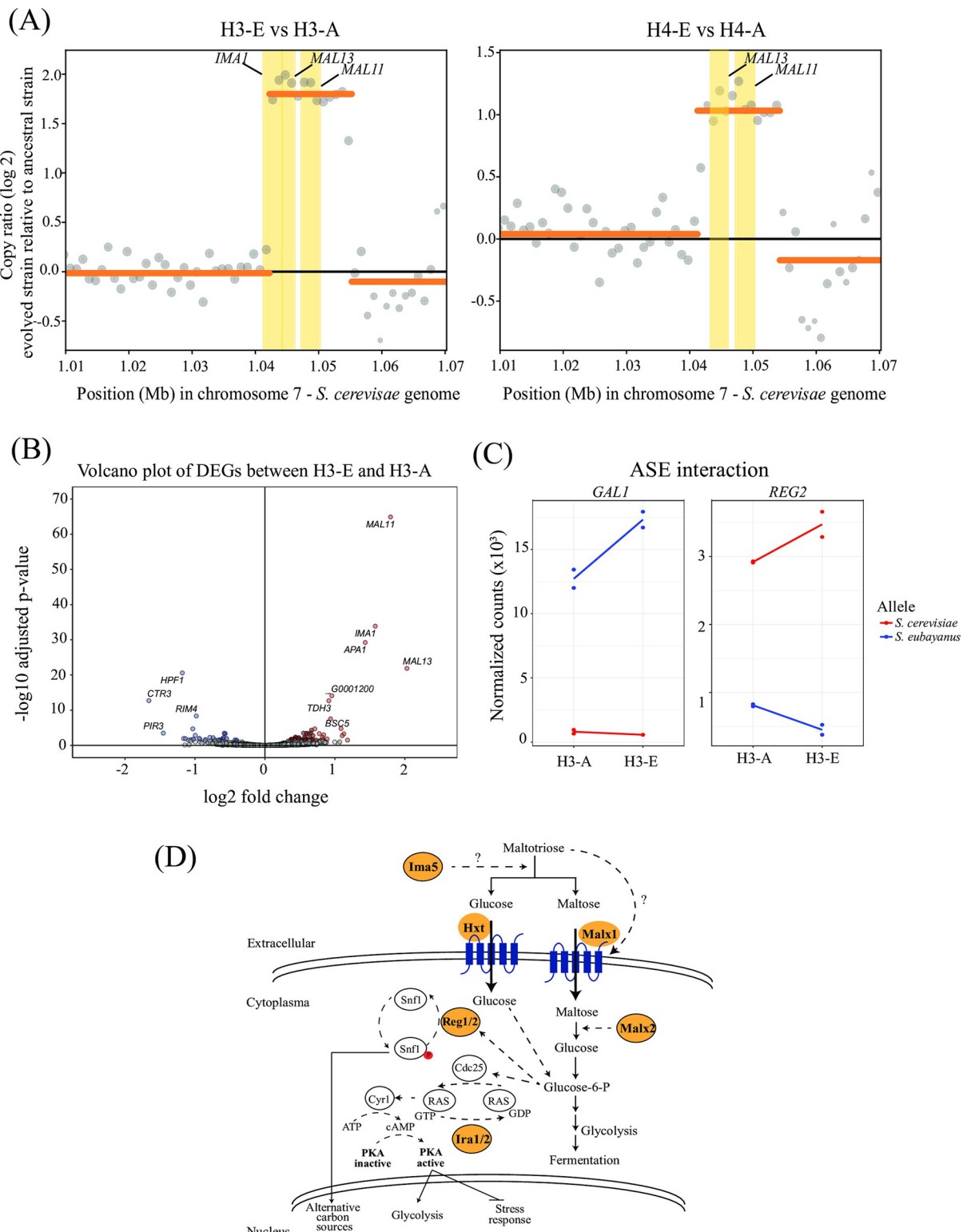

**Fig 5. Copy number variation and differential gene expression analysis.** (A) Copy number variations (CNVs) between H3-E and H4-E hybrids relative to their ancestral hybrids found in *S. cerevisiae* chromosome 7. Coding genes located within bins showing CNV calls higher than 1 copy (yellow rectangles) are shown. **(B)** Volcano plot showing differential expressed genes (DEGs) between H3-E and H3-A hybrids. The red and blue dots represent up-regulated and down-regulated genes in the H3-E hybrids, respectively. **(C)** Orthologous genes showing an interaction between allelic expression and experimental evolution. **(D)** Model depicting genes exhibiting mutations after the experimental

evolution assay (highlighted in orange) and involved in pathways related to the detection, regulation, uptake, and catabolism of maltotriose. Phosphorylation is indicated in red. In blue highlight transporters involved in sugar consumption.

16,69], opened new avenues for developing new strains to increase the diversity of fermentation profiles [11,70]. Previous studies using hybridization and experimental evolution demonstrated that lager yeast hybrids could be improved through selection under fermentation conditions [71] without polyploidization [72]. While these studies have expanded the diversity of lager yeast phenotypes, they are primarily based on a single *S. eubayanus* genetic background, CBS 12357T (belonging to PB-1), which is not representative of the overall species' genetic and phenotypic diversity [15,18,37,73]. *S. eubayanus* lineages vary widely in fermentation capacity and aroma profiles during beer fermentation, suggesting that the natural diversity of *S. eubayanus* is also well-suited for making innovative lager hybrid strains [18,44,73]. Here, we expanded the strain repertoire available for lager brewing by including all three different *S. eubayanus* lineages found in Patagonia. Leveraging the genetic diversity of *S. eubayanus*, we created novel *S. cerevisiae* x *S. eubayanus* hybrids and enhanced their fermentation capacity through experimental evolution. Interestingly, no polyploidies, aneuploidies, or loss of heterozygosity events were detected in our evolved hybrids, suggesting that these structural chromosomal changes might not be essential for efficient fermentation performance. Instead, we show that desirable phenotypic outcomes such as high ethanol production and new aroma profiles are the result of an intricate interplay of pre-existing genetic diversity, and selection on species-specific mitochondria, *de novo* mutations and differential expression in sugar consumption genes, together with CNV of the *MAL* genes, important to improve maltose consumption during fermentation.

Hybridization offers a mechanism to combine beneficial traits from different species, which can enable adaptation to new environmental conditions [3,74] and improve the yield of plant cultivars and animal breeds [8,75,76]. However, hybridization—without subsequent selection of desirable traits for multiple generations—may not be sufficient to generate new phenotypes. None of the initial F1 hybrids in our experiment showed best parent heterosis, demonstrating that hybridization at different temperatures alone, was not sufficient to generate hybrids with a greater fitness than their parents under fermentative conditions. We therefore turned to experimental evolution as an alternative approach to improve the hybrids' fermentative profiles. Experimental evolution across multiple generations, paired with time-series whole genome sequencing, is a powerful tool for studying microbial responses to a selective environment and understanding the fitness effects of *de novo* mutations [24–26,77]. We found that, after 250 generations in a high sugar and ethanol environment, hybrids evolved faster fermentation performance and higher ethanol production compared to both parents and ancestral unevolved hybrids. We also identified an evolved individual that produces significantly more ethanol than the commercial lager strain, likely due to differences in metabolic fluxes between them. For example, in the production of glycerol, acetic acid, and lactic acid, as well as in the consumption of more complex sugar sources. An alternative is to obtain improved strains by performing experimental evolution on parental strains, as previously described [44,78]. However, *S. eubayanus* has poor maltotriose consumption, a trait not easily improvable through artificial selection compared to commercial strains [79]. Instead, hybridization has the potential to produce more genetic variation than mutations alone. Thus, the hybrids combine traits of both parents, such as low-temperature tolerance and maltotriose uptake. In addition, the gene expression subgenomes crosstalk may generate different volatile compound combinations, increasing the variety of flavors and aromas [34]. Interestingly, the hybrids' evolutionary potential relied on the parental mitochondria. Our study shows that carrying mitochondria

from *S. eubayanus* provides a selective advantage under fermentation conditions and leads to greater evolutionary potential in hybrids. Consistent with our results, all lager commercial hybrids have *S. eubayanus* mitochondria [9,10,19]. It has been demonstrated that in synthetic hybrids, *S. eubayanus* mitochondria confers vigorous growth at colder temperatures compared to the *S. cerevisiae* mitotype, potentially conferring a competitive advantage in the cooler brewing conditions typical of lagers [31]. However, we performed experimental evolution at warmer temperatures (25°C). It is thus plausible that species-specific mitochondrial effects play an additional role, specifically concerning sugar utilization and glucose repression [80] when adapting to lager brewing conditions. These mitochondrial effects likely involve complex genetic interactions with the nuclear genome and might be exacerbated in the presence of the *S. eubayanus* mitochondria.

Our genome-wide screens for mutations to elucidate the genetic basis of hybrid fitness improvement identified *de novo* SNPs and CNVs in the genomes of the evolved lines and hybrids. These genetic changes were identified in genes with known effects on DNA replication and repair processes, maltose metabolism, and cell wall organization (**Fig 5D**). Particularly interesting is the mutation in *IRA2* in the *S. cerevisiae* subgenome which is related to carbon metabolism. Evolved hybrids carried a premature stop codon in the *IRA2* gene, absent in both the *S. cerevisiae* and *S. eubayanus* parental ancestors, and in the unevolved hybrids at the beginning of experimental evolution. Despite using different experimental conditions and hybrid genetic backgrounds, previous studies have also identified mutations in the *IRA2* gene in multiple evolved isolates, corroborating the important role of this gene in the fermentation process [21,29]. Null *ira2*$^{SC}$ mutants did not show the same phenotypes as the H3-E hybrid carrying the *IRA2* premature stop codon, suggesting a complex genetic interaction in this genetic background. We searched for genes with *de novo* mutations previously described to have genetic interactions with *IRA2* using the *Saccharomyces* Genome Database (SGD). Our search resulted in three genes potentially interacting with *IRA2*: *IMA1*, *MAL11*, and *CMC1*. *CMC1* encodes for a protein involved in the assembly of cytochrome c oxidase in the mitochondria [81] and contains a SNP in the corresponding *S. eubayanus* parental allele. In addition, *IMA1* and *MAL11* genes were up-regulated in the evolved hybrid and exhibited CNVs relative to the ancestral hybrid. Interactions between the *S. cerevisiae IRA2* allele with these genes could explain the differences observed between the *ira2*$^{SC}$ mutants and the H3-E hybrid. *IRA2* is required for reducing cAMP levels under nutrient limited conditions, where cAMP directly regulates the activity of several key enzymes of glycolysis [82,83]. A mutation in *IRA2* would increase the carbon flux through glycolysis, which is in agreement with our finding that evolved hybrids showed higher sugar consumption. Furthermore, the regulation of the yeast mitochondrial function in response to nutritional changes can be modulated by cAMP/PKA signalling [84], which might be exacerbated in strains carrying *S. eubayanus* mitochondria. We further consolidated this mechanism by CNV and transcriptome analyses, which detected several up-regulated genes related to maltose consumption in the evolved hybrid during fermentation. Fluctuations in chromosomal location and copy number of the *MAL* genes are present in many industrial strains [85] and *de novo* evolved hybrids [29], containing six or more copies of the *MAL3* locus. Furthermore, the newly generated hybrids exhibited a distinct volatile compound profile compared to the W34/70 lager strain. This highlights the potential of wild Patagonian yeast to introduce diversity into the current repertoire of available lager yeasts. Previous studies in laboratory-made lager hybrids revealed genetic changes that significantly impacted fermentation performance and changed the aroma profile of the resulting beer, compared to the commercial lager strain [22,71].

In summary, our study expands the genetic diversity of lager hybrids and shows that new *S. cerevisiae* x *S. eubayanus* hybrids can be generated from wild yeast strains isolated from

Patagonia. We found that hybridization at low temperatures, selecting for the retention of *S. eubayanus* mitochondria, followed by experimental evolution under fermentative conditions, and selection on desirable traits (ethanol production and aroma profiles), can generate hybrid strains with enhanced fermentation capacities. We delineate how genetic changes within distinct subgenomes of the hybrids contribute to improved fermentation efficacy, specifically in the context of cold lager brewing conditions. This opens up new opportunities for the brewing industry to alleviate current constraints in lager beer production and to expand the range of currently available lager beer styles.

## Supporting information

**S1 Fig. Generation of interspecific *S. cerevisiae* x *S. eubayanus* hybrids.** Experimental procedure designed to generate and identify interspecific hybrids at two different temperatures (12 and 20˚C).
(PDF)

**S2 Fig. Phenotypic characterization of *S. cerevisiae* parental strains.** (A) Fermentation performance of 15 *S. cerevisiae* strains. (B) Maximum OD reached of growth curves in maltotriose 2% under microculture conditions (C) Maltotriose uptake after growth in maltotriose 2% under microculture conditions. Plotted values correspond to three biological replicates. The (*) represents different levels of significance between the phenotype of haploid strains and their respective parental strain (t-test; $^*p \leq 0.05$, $^{**}p \leq 0.01$, $^{***}p \leq 0.001$, $^{****}p \leq 0.0001$ and ns: non-significant).
(PDF)

**S3 Fig. Fermentative capacity at 12˚C of each hybrid.** Each plot represents a different cross. The (*) represents different levels of significance between the phenotype of hybrids and their respective parental strain (t-test; $^*p \leq 0.05$, $^{**}p \leq 0.01$, $^{***}p \leq 0.001$, $^{****}p \leq 0.0001$).
(PDF)

**S4 Fig. Fitness comparison of evolved lines after 250 generations.** (A) Mean relative fitness (growth rate) of evolved lines after 250 generation under microculture conditions. Evolved lines were evaluated in the same medium where they were evolved (M o T medium). (B) Mean relative fitness (growth rate) comparison between evolved lines from hybrids generated at 12 and 20˚C. Plotted values correspond to the mean of three independent replicates of each evolved lines. The (*) represents different levels of significance between evolved lines and unevolved hybrid in (A) and from hybrids generated at 12˚C vs 20˚C in (B) (Students t-test, $^* p < 0.05$, $^{**} p < 0.01$, ns not significant).
(PDF)

**S5 Fig. Fitness dynamics of evolved lines in maltose and maltose with maltotriose.** (A) Mean relative fitness (growth rate and OD) of replicate population in 2% maltose. (B) Mean relative fitness (growth rate and OD) of replicate population in 1% maltose and 1% maltotriose. Plotted values correspond to the mean of three independent replicates of each evolved line.
(PDF)

**S6 Fig. Fitness dynamics of evolved lines in maltose and maltose with maltotriose under fermentation condition.** (A) Mean relative fitness (maximum $CO_2$ loss) of replicate population in 2% maltose. (B) Mean relative fitness (maximum $CO_2$ loss) of replicate population in 1% maltose and 1% maltotriose. Plotted values correspond to the mean of three independent

replicates of each evolved line.
(PDF)

**S7 Fig. Fermentative capacity of evolved individuals.** Fermentative capacity of evolved individuals relative to the commercial lager strain W34/70. Plotted values correspond to the mean of three independent replicates of each individual. The (*) represents different levels of significance between strains and commercial lager strain (Students t-test, * $p < 0.05$, ** $p < 0.01$, *** $p < 0.001$).
(PDF)

**S8 Fig. Dynamics of molecular evolution.** Allele frequencies over time in H3-4 and H4-1 lines evolved in T medium. In red and blue are highlighted SNPs in the genes present in the evolved individuals in the *S. cerevisiae* and *S. eubayanus* subgenome, respectively.
(PDF)

**S9 Fig. FACS analysis of H3-4 and H4-1 populations after 250 generations.** The number of cells versus propidium iodide intensity is shown. Haploid (n), diploid (2n), and tetraploid (4n).
(PDF)

**S1 Table.** (A) Strains used in this study. (B) Primers used in this study.
(XLSX)

**S2 Table.** (A) Phenotypic characterization of the *S. cerevisiae* strains under fermentation conditions (maximum $CO_2$ loss). (B) Statistical analysis of fermentative capacity of *S. cerevisiae* strains. (C) Sporulation efficiency and spore viability for *S. cerevisiae* and *S. eubayanus* strains.
(XLSX)

**S3 Table.** (A) Phenotypic characterization of the 31 interspecific hybrids and parental strains under microculture conditions. (B) Statistical analysis of phenotypes under microculture conditions. (C) Best-parent heterosis in the 31 interspecific hybrids evaluated under microculture conditions. (D) Mid-parent heterosis in the 31 interspecific hybrids evaluated under microculture conditions.
(XLSX)

**S4 Tables.** (A) Fermentation capacity (maximum $CO_2$ loss) of hybrids in 12˚Brix wort at 12˚C. (B) Statistical analysis of fermentative capacity of hybrids at 12˚C. (C) Best-parent heterosis for fermentative capacity. (D) Mid-parent heterosis for fermentative capacity. (E) Sugar consumption and ethanol production of four interspecific hybrids and parental strains. (D) Statistical analysis of maltotriose uptake and ethanol production.
(XLSX)

**S5 Table.** (A) Mean relative fitness (growth rate and OD) and statistical analysis of each of the evolved lines in maltose and maltose/maltotriose relative to unevolved hybrid. (B) Mean relative fitness (growth rate and OD) and statistical analysis of evolved hybrids in maltose and maltose/maltotriose relative to unevolved hybrid. (C) SNPs identified in the *COX3* gen. (D) Identity matrix derived from *COX3* gen multiple alignment.
(XLSX)

**S6 Table.** (A) Mean relative fitness and statistical analysis for maximum $CO_2$ loss of each of the evolved lines in maltose and maltose/maltotriose relative to unevolved hybrid. (B) Mean relative fitness and statistical analysis for maximum $CO_2$ loss of evolved hybrids in maltose and maltose/maltotriose relative to unevolved hybrid. (C) Maltotriose uptake and statistical

analysis of evolved lines in maltose and maltose/maltotriose relative to commercial lager strain W34/70. (D) Mean relative fitness and statistical analysis for maximum $CO_2$ loss of evolved lines in maltose and maltose/maltotriose relative to commercial lager strain W34/70.
(XLSX)

**S7 Table.** (A) Fermentative capacity, maltotriose uptake and ethanol production of evolved individuals of H3-A and H4-A hybrids. (B) Volatile compounds production of H3-4-C1 and W34/70 in beer wort.
(XLSX)

**S8 Table.** (A) Bioinformatics summary statistics. (B) Genomic contributions (%) from parental strains in the H3-E and H4-E hybrids. (C) LOH regions. (D) SnpEff analysis of the novel polymorphisms in H3-E, H4-E, and evolved population. (E) Number of SNPs in evolved H3-4 and H4-1 populations. (F) CNV results comparing evolved hybrids with their ancestral hybrid. Only CNVs with 1 or more copies are listed. (G) RNA-seq analysis between H3-E and H3-A hybrids. (H) Enriched GO terms of hybrid genes showing differential expression between ancestral and evolved hybrids. (I) Genes exhibiting Allele-Specific Expression (ASE), with values approximating 1 indicating overexpression of *S. cerevisiae* alleles, and values close to 0 representing overexpression of *S. eubayanus* alleles.
(XLSX)

**S9 Table.** Fermentative capacity and maltotriose uptake of *ira2* mutants.
(XLSX)

## Acknowledgments

We thank Antonio Molina, José Ruiz, Kamila Urbina, Mirjam Amcoff, Elin Gülich and S. Lorena Ament-Velásquez for their technical help. We also acknowledge Fundación Ciencia & Vida for providing infrastructure, laboratory space and equipment for experiments. Additionally, we acknowledge the use of the server provided by the supercomputing infrastructure of the National Laboratory for High Performance Computing Chile (NLHPC, ECM-02) and by SNIC through Uppsala Multidisciplinary Center for Advanced Computational Science (UPPMAX) under Project naiss2023-22-62.

## Author Contributions

**Conceptualization:** Jennifer Molinet, Francisco A. Cubillos.

**Data curation:** Jennifer Molinet, Carlos A. Villarroel, Pablo Villarreal.

**Formal analysis:** Jennifer Molinet, Carlos A. Villarroel, Pablo Villarreal, Roberto F. Nespolo.

**Funding acquisition:** Roberto F. Nespolo, Rike Stelkens, Francisco A. Cubillos.

**Investigation:** Juan P. Navarrete, Pablo Villarreal, Felipe I. Sandoval, Roberto F. Nespolo, Francisco A. Cubillos.

**Methodology:** Juan P. Navarrete, Carlos A. Villarroel, Pablo Villarreal, Felipe I. Sandoval.

**Resources:** Roberto F. Nespolo, Rike Stelkens.

**Supervision:** Francisco A. Cubillos.

**Validation:** Francisco A. Cubillos.

**Writing – original draft:** Jennifer Molinet, Rike Stelkens, Francisco A. Cubillos.

**Writing – review & editing:** Jennifer Molinet, Pablo Villarreal, Rike Stelkens, Francisco A. Cubillos.

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
