## [Decision Letter · Decision Letter 0]

25 Mar 2024

Dear Dr Cubillos,

Thank you very much for submitting your Research Article entitled 'Wild Patagonian yeast improve the evolutionary potential of novel interspecific hybrid strains for Lager brewing' to PLOS Genetics.

The manuscript was fully evaluated at the editorial level and by independent peer reviewers. The reviewers, especially reviewer #1, raised some important questions and concerns about the current manuscript. Based on the reviews, we will not be able to accept this version of the manuscript, but we would be willing to review a much-revised version. We cannot, of course, promise publication at that time.

Moreover, in line 59 you write "An iconic example is the domestication of the hybrid yeast Saccharomyces pastorianus”. I wonder what you exactly mean by domestication here as the hybrid is itself the domesticate and therefore in the case of S. pastorianus, domestication corresponds to hybridization.

If you decide to revise the manuscript for further consideration at PLOS Genetics, please aim to resubmit within the next 60 days, unless it will take extra time to address the concerns of the reviewers, in which case we would appreciate an expected resubmission date by email to plosgenetics@plos.org.

We are sorry that we cannot be more positive about your manuscript at this stage. Please do not hesitate to contact us if you have any concerns or questions.

Yours sincerely,

José Paulo Sampaio

Guest Editor

PLOS Genetics

Geraldine Butler

Section Editor

PLOS Genetics

Reviewer's Responses to Questions

**Comments to the Authors:**

Reviewer #1: Molinet et al seek to generate novel S. cerevisiae x S. eubayanus hybrids to increase genetic and phenotypic diversity of lager brewing strains. They leverage natural genetic diversity of S. eubayanus strains from Chile, cross to S. cerevisiae wine strains, and phenotype hybrids for a variety of fermentation related traits. They select three hybrids to evolve under maltose + ethanol and maltose/maltortriose + ethanol conditions, evolve for 250 generations, and phenotype evolved hybrid populations and clones. They sequence the genomes of two clones, and identify a number of point mutations and copy number changes associated with sugar consumption. They find S. eubayanus mitochondria to be important in certain fermentation traits beyond the previously recognized effect on temperature. They use RNA sequencing and find increased expression of maltose genes in evolved lines and also demonstrate a role for IRA2 mutations in increasing fermentation capacity. A number of unique insights result from this work, particularly highlighting mechanisms involved in sugar sensing and catabolism.

While I think the motivation behind this study to explore greater genetic and phenotypic diversity in lager hybrids through novel hybridization is exciting and interesting, I feel there are some additional analyses and discussion that could be addressed to strengthen the manuscript.

First, it was unclear to me why the authors did not sequence additional clones or populations from the evolved lines. While the two clones they did sequence revealed important genetic changes that may underlie phenotypes of interest, it is difficult to assess generalizable findings. If a goal of this paper is to understand the genetic changes underlying increased fermentation capacity, by exploring the mutations that occurred across different replicates and genotypes, the authors would have more power to conclude what changes are necessary/sufficient and repeatable.

Second, since one of the stated motivations for this study is to better understand lager domestication/evolution, and since one could frame this study as a sort of lager domestication experiment, it has great potential to better understand the changes that lead to modern day lager strains. What are commonalities or differences between lager strains and evolved hybrids from this study? I understand no aneuploidy events were identified in the selected evolved hybrids (but see comment above about only sequencing two clones), but did the authors look for ploidy changes or loss of heterozygosity events? Are any of the identified mutations present in previously sequenced beer strains? This would be really interesting to discuss further, as the lager strains have quite complex karyotypes, and if the authors can recapitulate a lot of lager fermentation traits without ploidy, aneuploidy, and LOH, this could suggest that many of those changes are not necessary for lager fermentation (although the cold phenotype is not assessed here, so that context may be less clear).

Third, I might suggest a stylistic choice to condense some of the phenotypic results presented in Figures 1 and 2, and the associated text, in order to highlight the insights from the evolution experiments and the genomic, genetic, and transcriptomic results, which are quite interesting and feel a little de-emphasized in the manuscript.

Minor comments

Can the authors comment on their S. cerevisiae strain selection choice? Did they consider using a beer strain?

Several of the figures were a bit challenging for me to understand. For example in Figures 2-4, I think clarity could be improved by including more text on the figures and in the figure legend to demonstrate which population is being compared to what (statistics, relative growth compared to other evolved lines or to the ancestor, etc). For Figure 2 - Is the only theorized difference between the hybridizations at 12 or 20 which species mtDNA they inherit? If so, I suggest changing the figure legend from hybridization temperature to S. cerevisiae mtDNA and S. eubayanus mtDNA.

Please provide additional information and context for the volatile analysis. Are the compounds identified in the novel hybrids pleasant or aversive? Are there changes in volatiles between ancestor and evolved hybrids? The authors note the presence of 4VG, which is unfavorable in many styles of beer including lager. The authors mention several times that one of the motivations for new lager hybrids is to generate novel aromas/flavors, but it is unclear if the compounds found here would be desirable or not, and what the authors think would be needed to address this.

Can the authors comment on the number of SNPs identified between the two hybrids? Do they suspect a mutator allele in the one evolved line?

A little more explanation about the ira2 knockout creation/results would be helpful. Is the ira2 knockout in H3E just a total deletion of the ira2 gene in the evolved hybrid background that already has a IRA2 premature stop codon? Since the premature stop has a different phenotype than the null, the authors should be careful with the language about this result, since they did not test the effect of the premature stop codon. This finding about complex interactions is very interesting. Since this mutation occurred early in the experiment, were there any other de novo mutations in this background that could explain this interaction?

Reviewer #2: See attached comments.

Reviewer #3: The manuscript by Molinet et al. describes the generation of new S. cerevisiae x S. eubayanus hybrids and their improvement through experimental evolution. The evolved hybrids outperform the ancestral hybrids and the parental strains in wort fermentations. Whole genome sequencing and RNAseq are used to identify genetic changes in the evolved hybrids, which might explain the enhanced fermentation. The study is technically sound, and the manuscript is well written and is a useful contribution to the yeast and brewing community. I only have a couple of comments and suggestions for improving it further:

General comment: you use experimental evolution on the hybrids to improve the fermentation capacity beyond the parental strains. Isn't it likely that one could have seen a similar improvement by performing the experimental evolution on the parental strains directly (e.g. we saw this in dx.doi.org/10.1128/AEM.02302-17)? You could maybe comment on this in the discussion, and also explain what the potential benefits of including the hybridization are.

Throughout. 'Lager' -> 'lager'

Line 38. greater

Lines 125-127. What is the genetic background of these strains? Are any of them brewing strains (Ale beer or Mosaic beer) or isolated from brewing environments?

Line 332. Maybe "such as growth on glucose, maltose, maltotriose, and in the presence of ethanol or sorbitol"?

Figure 1. Could you clarify what the parameters 'Diauxic shift (lag time)' and 'Diauxic shift (umax)' are?

Figure 3A and B. Shouldn't mass loss as CO2 and ethanol formation be strongly correlated? Here the strain with the highest ethanol formation didn't have the highest CO2 produced. Maybe worth double-checking the numbers.

Line 468. 3.6 - 3.8% ABV is quite low for a 12P wort. Was there a lot of unfermented sugars still in the beers?

Figure 3C and Lines 484-491. In the heatmap labels you have fatty acids (e.g. hexanoic acid), but in the text you talk about ethyl esters of these fatty acids. Octanoic acid is also present twice in the heatmap. In the heatmap, is 3-methylbutyl ester 3-methylbutyl acetate? In the text you use 4-vinyl guaiacol, while in the heatmap you have 2-methoxy-4-vinylphenol.

Line 531 instead of "CRISPR assay", maybe just "CRISPR-Cas9 gene editing"?

KK

**Have all data underlying the figures and results presented in the manuscript been provided?**

Reviewer #1: Yes

Reviewer #2: Yes

Reviewer #3: Yes

PLOS authors have the option to publish the peer review history of their article (what does this mean?). If published, this will include your full peer review and any attached files.

Reviewer #1: No

Reviewer #2: No

Reviewer #3: No

---

## [Decision Letter · Decision Letter 1]

17 May 2024

Dear Dr Cubillos,

We are pleased to inform you that your manuscript entitled "Wild Patagonian yeast improve the evolutionary potential of novel interspecific hybrid strains for lager brewing" has been editorially accepted for publication in PLOS Genetics. Congratulations!

Reviewer #1 suggested adding a sentence in the discussion concerning the limited number of evolved populations sequenced and I also think that it might improve this part.

Yours sincerely,

José Paulo Sampaio

Guest Editor

PLOS Genetics

Geraldine Butler

Section Editor

PLOS Genetics

Comments from the reviewers (if applicable):

Reviewer's Responses to Questions

**Comments to the Authors:**

Reviewer #1: The authors addressed all my comments with additional analyses and revisions to the text. Thanks to the authors for their thoughtful responses. The one comment I maintain is that additional sequencing would be beneficial, but I understand that budget limitations were prohibitive for this. I suggest the authors add a sentence to the discussion acknowledging that sequencing was only done for two evolved populations, and that additional insight may be gleaned from future work sequencing other populations (or something to this effect).

Reviewer #2: Comments uploaded

Reviewer #3: The authors have addressed my concerns.

**Have all data underlying the figures and results presented in the manuscript been provided?**

Reviewer #1: Yes

Reviewer #2: Yes

Reviewer #3: Yes

PLOS authors have the option to publish the peer review history of their article (what does this mean?). If published, this will include your full peer review and any attached files.

Reviewer #1: No

Reviewer #2: No

Reviewer #3: No

**Data Deposition**

http://datadryad.org/submit?journalID=pgenetics&manu=PGENETICS-D-24-00109R1

**Press Queries**

---

## [Editor Report · Acceptance letter]

24 May 2024

PGENETICS-D-24-00109R1 

Wild Patagonian yeast improve the evolutionary potential of novel interspecific hybrid strains for lager brewing 

Dear Dr Cubillos, 

We are pleased to inform you that your manuscript entitled "Wild Patagonian yeast improve the evolutionary potential of novel interspecific hybrid strains for lager brewing" has been formally accepted for publication in PLOS Genetics! Your manuscript is now with our production department and you will be notified of the publication date in due course.

With kind regards,

Zsofia Freund

PLOS Genetics

On behalf of:
